# Diagnostic, Predictive and Prognostic Molecular Biomarkers in Pancreatic Cancer: An Overview for Clinicians

**DOI:** 10.3390/cancers13051071

**Published:** 2021-03-03

**Authors:** Dimitrios Giannis, Dimitrios Moris, Andrew S. Barbas

**Affiliations:** 1Institute of Health Innovations and Outcomes Research, The Feinstein Institutes for Medical Research, Northwell Health, Manhasset, NY 11030, USA; dgiannis@northwell.edu; 2Department of Surgery, Duke University Medical Center, Durham, NC 27710, USA; andrew.barbas@duke.edu

**Keywords:** pancreatic cancer, pancreatic ductal adenocarcinoma, biomarkers, predictive markers, prognostic markers

## Abstract

**Simple Summary:**

Pancreatic cancer is the fourth most common cancer-related cause of death in the United States and is usually asymptomatic in early stages. There is a scarcity of tests that facilitate early diagnosis or accurately predict the disease progression. To this end, biomarkers have been identified as important tools in the diagnosis and management of pancreatic cancer. Despite the increasing number of biomarkers described in the literature, most of them have demonstrated moderate sensitivity and/or specificity and are far from being considered as screening tests. More efficient non-invasive biomarkers are needed to facilitate early-stage diagnosis and interventions. Multi-disciplinary collaboration might be required to facilitate the identification of such markers.

**Abstract:**

Pancreatic ductal adenocarcinoma (PDAC) is the most common pancreatic malignancy and is associated with aggressive tumor behavior and poor prognosis. Most patients with PDAC present with an advanced disease stage and treatment-resistant tumors. The lack of noninvasive tests for PDAC diagnosis and survival prediction mandates the identification of novel biomarkers. The early identification of high-risk patients and patients with PDAC is of utmost importance. In addition, the identification of molecules that are associated with tumor biology, aggressiveness, and metastatic potential is crucial to predict survival and to provide patients with personalized treatment regimens. In this review, we summarize the current literature and focus on newer biomarkers, which are continuously added to the armamentarium of PDAC screening, predictive tools, and prognostic tools.

## 1. Introduction

Pancreatic cancer is the fourth most common cancer-related cause of death in the United States [1]. The median age of diagnosis is 70 years, and only up to 10% of reported cases occur prior to the fifth decade of life [2]. Over the last decade, the reported incidence of pancreatic cancer was 13.1 per 100,000 men and women per year [3]. According to the Surveillance, Epidemiology, and End Results (SEER) database, approximately 47,000 people will die due to pancreatic cancer, representing 7.8% of all cancer-related deaths, and over 55,000 new cases were expected in the United States in 2020. The highest incidence reported in USA is 16.9 per 100,000 persons in Black men, while males have shown a higher incidence compared to females (14.9 vs. 11.6 per 100,000 persons, respectively) [3].

Clinical presentation depends on tumor location, and most pancreatic ductal adenocarcinoma (PDAC) tumors affect the pancreatic head. Patients present with obstructive jaundice and subtle gastrointestinal symptoms, such as vague abdominal discomfort or nausea, and systemic manifestations at advanced stage disease (weight loss, cachexia and anorexia). Other findings include the dysregulation of blood glucose levels and less common presentations, such as duodenal obstruction or bleeding [4]. Early-stage PDAC is usually asymptomatic, and patients presenting with clinical manifestations commonly have advanced, non-resectable tumors [5].

After diagnosis, the overall survival ranges between four and six months, with a five-year survival rate of <5%. However, survival is significantly higher in early stage disease (IA: 31.7%; IB: 11.8%) compared to more advanced stages (IIA: 9.0%; IIB: 8.7%; III: 1.9%; IV: 0.5%) [6]. This survival disparity, even in early stages (IA vs IB) makes early diagnosis one of the most important aspects in the treatment of pancreatic cancer. However, early diagnosis is difficult due to the absence of symptoms and adequate screening tests.

Biomarkers are important tools to diagnose cancer, to determine prognosis, and to select appropriate treatment. Biomarkers are detectable in plasma, urine, and saliva specimens, and they offer a non-invasive, relatively inexpensive, and efficient monitoring method [7]. Currently, there are a lack of biomarkers that can be detected in the time interval between carcinogenesis and invasion, when the disease is potentially curable, due to low sensitivity and specificity in early-stage small tumors [8].

In this review, we summarize the current literature on the molecular biomarkers in pancreatic cancer that have potential roles in clinical practice (Figure 1 and Table 1).

## 2. Biomarkers 

### 2.1. CA 19-9: Diagnostic and Prognostic Role

Carbohydrate antigen 19-9 (CA 19-9) has been classically used in the diagnosis of pancreatic cancer, and its usefulness has been extensively evaluated [131]. Goonetilleke et al., in their systematic review, assessed the median sensitivity and specificity of CA 19-9 in symptomatic patients and reported values of 79% and 82%, respectively [9]. 

The composite marker of increased CA 19-9, weight loss, and hyperbilirubinemia had 100% specificity and positive predictive value in patients undergoing surgery for suspected pancreatic malignancy [10]. Van Manen et al. reported a 91.4% positive predictive value of combined carcinoembryonic antigen (CEA) (>7.0 ng/mL) and CA 19-9 (>305 U/mL) in predicting the presence of advanced PDAC [11].

The preoperative elevation of CA 19-9 in patients with resectable tumors has been associated with a decreased overall survival [12]. Both postoperative decreases in serum CA 19-9 and postoperative CA 19-9 < 200 U/mL are strongly associated with survival. In addition, preoperative CA 19-9 levels are lower in patients without nodal disease (N0) and an earlier stage of disease compared to patients with positive nodes or an advanced disease, respectively [13]. Berardi et al. reported a significantly higher overall survival in patients who received chemotherapy for locally advanced or metastatic pancreatic cancer and had CA 19-9 ≤ 37 U/mL compared to patients with CA 19-9 > 37 U/mL (18.49 vs. 9.21 months, respectively) [14].

Serum CA 19-9 is not useful as a screening tool in asymptomatic patients due to its low positive predictive value that ranges between 0.5 and 0.9% [15,16]. CA 19-9 levels may increase in nonmalignant hepatobiliary and pancreatic diseases such as acute cholangitis, acute pancreatitis, acute liver cirrhosis as well as malignancies such as cholangiocarcinoma, ovarian cancer, and colorectal cancer [17,18,19,20,21,22]. In addition, 5–10% of the population are negative for the sialyl Lewis epitope, and, if affected by pancreatic cancer, CA 19-9 (sialyl Lewis a) is absent and of no diagnostic use in these individuals [23,24].

### 2.2. microRNAs: Diagnostic and Prognostic Role

MicroRNAs (miRNA) are non-coding RNAs that regulate gene expression at the post-transcriptional level through interaction with target mRNAs at the 3′ untranslated region (3′ UTR) and the induction of mRNA degradation [25]. These molecules have been detected in various body fluids, such as serum, saliva, breast milk/colostrum, urine, and peritoneal cavity fluid, and they have recently emerged as potential cancer biomarkers [25,26].

Bloomston et al. showed that PDAC tissue has a distinct miRNA expression profile that can distinguish it from normal pancreatic tissue and chronic pancreatitis. Furthermore, they identified a group of six miRNAs (miR-452, miR-105, miR-127, miR-518a-2, miR-187, and miR-30a-3p) that could classify patients with nodal disease into long-term (>24 months) or short-term survivors (<24 months). In the same study, miR-196a-2 was found to be associated with a poor survival [27]. 

Yang et al. reported a similar pattern in miRNA derived from pancreatic fluid/tumor tissue and stool between patients with PDAC versus chronic pancreatitis or normal pancreatic tissue. PDAC patients had higher miR-21 and miR-155 and lower mir-216 expression compared to chronic pancreatitis or normal controls [28].

Urine miRNAs have also demonstrated efficacy as non-invasive biomarkers of PDAC. Patients with a stage I disease predominantly express miR-143, miR-223, and miR-30e at higher levels than the healthy population and miR-143, miR-223 and miR-204 at higher levels than patients with a stage II–IV disease. In addition, miR-223 and miR-204 have been reported to distinguish between early stage PDAC and chronic pancreatitis [29].

### 2.3. DNA Methylation Patterns: Diagnostic and Prognostic Role

DNA methylation is a process mediated by the DNA methyltransferase (DNMT) enzymes that results in the modification of cytosine residues through the addition of a methyl side group and the formation of 5-methylcytosine [30,31]. DNA methylation is a critical process that promotes the development of malignant cells through the activation of oncogenes and the inactivation of tumor suppressor genes [32]. The DNA methylation pattern in cfDNA, pancreatic secretions, or brush samples is a biomarker with significant potential due to the early occurrence of this modification in carcinogenesis, the long-term preservation of methylated DNA molecules in fixed samples, its widespread presence in tissues and body secretions, and its cell and tissue specificity [31]. The methylation of genes *CD1D* and *NDRG4* identified in pancreatic secretions has been associated with approximately 70% or higher sensitivity and 90% specificity in terms of discrimination between pancreatic cancer and normal pancreas or chronic pancreatitis [33]. Matsubayashi et al. reported a sensitivity of >90% for the detection of pancreatic cancer in patients with methylated *NPTX2* and *SPARC* genes in pancreatic secretion-derived DNA, whereas the methylation of *Cyclin D2* and *SARP2* genes demonstrated a sensitivity of approximately 82% [34]. Similarly, Parsi et al. investigated the presence of three methylated genes (cyclin D2, NPTX2, and TFPI2) in endoscopic brush samples and reported a positivity of 73.2% in PDAC patients compared to 13.6% in patients with benign biliary disease [35].

An analysis of promoter methylated genes in the circulating cfDNA of PDAC patients revealed a significantly decreased survival in patients with >10 hypermethylated genes compared to patients with <10 hypermethylated genes [36]. Specifically, the hypermethylation of *SFRP1, BMP3*, and *TFPI2* detected in cfDNA has been significantly associated with poor prognosis in stage IV PDAC patients. The aforementioned genes are involved in multiple pathways and cellular functions that affect carcinogenesis, including the Wnt pathway (*SFRP1),* the TGF-β pathway (*BMP3*), and cell adhesion (*TFPI2*) [36]. 

### 2.4. Mismatch Repair Genes and Microsatellite Instability: Diagnostic and Predictive Role

Microsatellites are short sequences (up to six base pairs) that are present in repetitive patterns throughout the genomic DNA. The deficient function of the mismatch repair proteins (MMRs), which identify and repair errors in DNA base insertion or deletion, results in mutations that affect the integrity of microsatellite DNA [37]. Cancers deficient in MMR are significantly less likely to have mutations in usual pancreatic cancer genes such as KRAS and mothers against decapentaplegic homolog 4 (SMAD4) but are more likely to have mutations in genes that generate cancers with microsatellite instability like ACV2RA and JAK1 [38]. Microsatellite instability has recently emerged as a potential biomarker associated with both diagnostic and predictive applications. Pancreatic malignancies have microsatellite instability at a frequency of 0.5–1% and a deficiency of mismatch repair proteins in 0.8–1.6% [37,39,40,41,42]. Interestingly, microsatellite instability has been recently associated with a mucinous PDAC histologic subtype [41]. 

Patients with PDAC and microsatellite instability had been reported to have a significantly prolonged survival in the presence of microsatellite instability compared to patients without mismatch repair deficiency (62 months vs. 10 months, respectively; *p* = 0.011) [43]. In contrast, Lupinacci and colleagues investigated 445 PDAC specimens and reported similar median disease-free (21.4 months vs. 15.6 months; *p* = 0.703) and overall survival rates (35.1 vs. 29.2; *p* = 0.663) between patients with and without microsatellite instability [41]. Ottenhof et al. analyzed 78 PDAC specimens and reported that the expression of mismatch repair proteins (MLH1, MSH2, MSH6, and PMS2) was not associated with prognosis [44]. 

The KEYNOTE clinical trial demonstrated a benefit of the immune checkpoint inhibitor pembrolizumab (objective response rate: 34.3%) in patients with unresectable or metastatic non-colorectal malignancies, including pancreatic tumors, with microsatellite instability identified in biopsy specimens [45]. In addition, PDAC patients with metastatic disease and mismatch repair gene mutations have been reported to have promising results in terms of the median overall survival (16.5 months) with systemic chemotherapy treatment [46]. In contrast, Riazy et al. reported a survival benefit in PDAC patients with adequate mismatch repair functionality treated with 5-fluorouracil or gemcitabine, while there was no difference between treated and untreated mismatch repair deficient patients [47].

### 2.5. KRAS Mutations: Diagnostic and Prognostic Role

Mutations of the *KRAS* gene are commonly identified in PDAC tumors (~90%) and represent an early molecular event in the pathogenesis of pancreatic intraepithelial neoplasias and adenocarcinomas by being involved in tumor progression and maintenance [39,48,49,50,51,52]. *KRAS* mutations are identified in both fine-needle aspiration-derived tissue DNA and circulating cfDNA from patients with PDAC, with a concordance of 77.3% between the detection methods [53]. Patients with the *KRAS* mutation in cfDNA were found to have a significantly shorter survival compared to patients without *KRAS* mutations [53]. *KRAS* mutations detected in PDAC tissue samples have been associated with the poorer survival of patients across all stages [54]. Specific *KRAS* mutations (Kras^G12D^ and Kras^G12V^) have been identified as independent survival prognostic markers, and the Kras^G12V^ mutation has been associated with increased levels of regulatory T cells and worse prognosis [55]. Lee et al. reported that the detection of circulating tumor cfDNA (identified through *KRAS* gene mutations), pre- and post-operatively, was associated with an increased risk of disease recurrence and a significantly worse overall survival compared to PDAC patients with non-detectable cfDNA [56]. Recently, a KRAS imbalance was linked to disease stage, with major imbalance favoring mutant KRAS expression to be found in metastatic rather than primary pancreatic tumors [57].

### 2.6. Exosomes: Diagnostic and Prognostic Role

Exosomes are extracellular vesicles derived from cells, and their diameter ranges between 50 and 150 nm. Cargo within endosomes includes a variety of biomarkers, such as disease-specific RNA molecules, proteins, DNA, and signaling molecules. Exosomes mediate both local and remote intercellular communication processes. Their role is important in pathophysiologic processes, including inflammation and cancer growth and spread, through angiogenesis, stromal cell activation, extracellular matrix remodeling, and immunosuppression [58,59].

Exosomes are currently being investigated as screening and diagnostic biomarkers due to their high-level secretion and circulation in patients with cancer and their tumor-specific cargo (disease-specific RNAs and proteins). Cancer cells secrete 10× more exosomes in comparison to normal cells, and an analysis of tumor-derived exosomes can provide important tumor profile related information [7]. These extracellular vesicles have been utilized as potential diagnostic markers in various malignancies (breast cancer, lung cancer, colorectal cancer, and hepatocellular carcinoma) [60,61,62,63]. 

In patients with PDAC, circulating exosomes can be used to identify specific DNA mutations, to determine prognosis, and to select appropriate treatment modalities. DNA mutations involved in PDAC pathogenesis (*KRAS* and *TP53*) are detectable in exosomal DNA derived from PDAC patient serum and in a higher frequency compared to that reported for KRAS mutations detected in cfDNA. Nevertheless, circulating mutant *KRAS* has been detected in healthy tissue samples, and the liquid biopsy findings should be carefully interpreted [64,65].

Glypican-1 (GPC-1) is a protein previously reported to be overexpressed in PDAC and plays an important role in signal transduction initiated by mitogenic molecules (HB-EGF, FGF-2 and TGF-β) [66,67,68]. Glypican is a heparan sulfate proteoglycan that is connected to the cell membrane through a glycosyl-phosphatidylinositol (GPI) anchor [59]. GPC-1 overexpression has been associated with perineural invasion in PDAC and is an independent prognostic factor of worse survival [69]. Exosomal GPC-1 has recently emerged as a marker with the potential to detect early stage PDAC and differentiate between benign and malignant pancreatic disease [70]. GPC-1+ exosomes levels are correlated with tumor burden and are associated with survival. An analysis of GPC-1+ exosomes in mice with PDAC showed that it could effectively detect intraepithelial lesions, even in the presence of negative MRI [7,70].

Exosomal miRNA has also been investigated as a potential biomarker of PDAC. Exosomal miR-17-5 has been found in higher levels in PDAC patients and has been associated with the progression, advanced stage, and metastasis of PDAC [71]. In addition, serum exosomal miR-1246, miR-3976, miR-4306, and miR-4644 as well as salivary exosomal miR-1246 and miR-4644, have been identified at higher levels in PDAC patients [72].

### 2.7. Circulating Tumor Cells: Diagnostic and Prognostic Role

Circulating tumor cells (CTCs) have been evaluated as diagnostic and survival prognostic markers in patients with PDAC [73]. Peripheral CTCs represent cells originating from the primary lesion that may be undetectable by imaging tests and non-accessible to biopsy with imaging guidance [74]. CTCs have been identified by cytology or KRAS mutation detection in PDAC patients with localized, locally invasive, and metastatic tumors [75]. Nevertheless, CTC isolation is a challenging process, and there is a high variability in CTC detection methods [75]. A recent meta-analysis by Zhu et al. investigated the diagnostic role of CTCs in PDAC and revealed a pooled sensitivity of 74% and a pooled specificity of 83% [76]. A meta-analysis by Wang et al. revealed that CTC-positive patients have shorter overall survival and progression-free survival rates compared to CTC-negative patients [77].

### 2.8. PAM4/MUC5AC: Diagnostic Role

PAM4 (clivatuzumab) is a monoclonal antibody with a high specificity for PDAC detection, even at early stages, and a high discriminatory ability between normal or benign pancreatic tissue and early stage PDAC, including pancreatic intraepithelial or intraductal mucinous neoplasms [78]. PAM4 reacts with the epitope mucin 5AC (MUC5AC), which is a highly expressed and secreted mucin of PDAC, and can be used in enzyme-linked immunoassays to detect early stage PDAC [79,80]. The overall sensitivity for PDAC detection is up to 76%, with 64% and 85% sensitivity in patients with early and advanced stage diseases, respectively. The combination of PAM4 and CA 19-9 previously achieved an overall sensitivity 84% and specificity of 82% [78]. PAM4 has been reported to effectively differentiate between PDAC and chronic pancreatitis, with only 19% of chronic pancreatitis specimens staining positively and most of their reactivity attributed to coexistent pancreatic intraepithelial neoplasia [81].

### 2.9. Osteopontin: Diagnostic and Prognostic Role

Osteopontin is an extracellular matrix-associated phosphoprotein that is normally produced by macrophages, osteoblasts, vascular smooth muscle cells, and endothelial cells, and it is detected in various body secretions [82,83]. Plasma osteopontin levels were initially reported to increase as much as 2.5× in patients with PDAC compared to normal controls. Osteopontin levels >334 ng/mL have been associated with a sensitivity of 80% and a specificity of 97% in the detection of PDAC, while the combination of CA 19-9 > 70 units/mL and osteopontin >334 ng/mL demonstrated a 100% sensitivity in PDAC detection [82]. Osteopontin levels can effectively differentiate PDAC patients from patients with chronic pancreatitis and healthy individuals, and a combination of osteopontin, CA 19-9, and TIMP-1 is superior to CA 19-9 or osteopontin alone in PDAC diagnosis [84,85]. In addition, osteopontin levels > 150 ng/mL have been associated with a shorter survival in patients with PDAC [84].

### 2.10. SMAD4/DPC4: Diagnostic, Prognostic and Predictive Role

The SMAD4 or DPC4 (deleted in pancreatic cancer-4) is a signal transduction protein that acts as the central mediator in the TGF-β signal transduction pathway [86]. The SMAD4 gene has been identified as one of the most important tumor suppressor genes involved in the development of late-stage pancreatic intraepithelial neoplasia in the pathogenesis of PDAC, epithelial-to-mesenchymal transition, and metastasis, with approximately 55% of PDAC tumors being affected by SMAD4-inactivating mutations [50,87,88,89,90,91]. Needle or core biopsy specimens stained with immunohistochemistry for SMAD expression facilitate the discrimination between PDAC and benign or inflammatory conditions [92].

The expression of the SMAD4 protein has been positively associated with an increased survival and negatively associated with the grade of intraepithelial lesions [87,93]. Interestingly, SMAD4 deletion has been associated with a shorter disease-free survival without affecting the overall survival [94]. A meta-analysis by Shugang and colleagues revealed an increased risk of death in PDAC patients with SMAD4 deletion, in both univariate (hazard ratio: 1.20; 95% CI: 1.03–1.40) and multivariate analysis (hazard ratio: 1.88; 95% CI: 1.31–2.70) data pooling [95].

In vitro and in vivo studies have demonstrated a role of SMAD deletion in the promotion of radioresistance in PDAC cells through the activation of autophagy and the clearance of radiation-induced cytotoxic oxidative products [96]. Notably, decreased SMAD expression results in an increased sensitivity of PDAC to drugs that target the cell cycle, including gemcitabine and cytarabine [94].

### 2.11. Immune Response and Inflammatory Markers: Prognostic and Predictive Role

Inflammation is an important component of both the progression of carcinogenesis and antitumor response. Inflammatory markers undergo various changes that are reflected in core laboratory parameters. Schlick et al. reported that the C-reactive protein and the neutrophil/lymphocyte (NLR) ratio are independent prognostic factors of poor survival in patients with PDAC. These markers were lower at the time of PDAC diagnosis in patients eventually needing second or third line chemotherapeutic agents and have been proposed as prognostic markers of poor response to chemotherapy [97]. In a study by Hoshimoto and colleagues, patients with high preoperative or postoperative platelet/lymphocyte (PLR) and NLR ratios had a significantly shorter overall survival [98]. Similarly, in another study by Giakoustidis and colleagues, both pretreatment (prior to surgery or prior to chemo/chemoradiotherapy) NLR > 4 and PLR > 120 were associated with a shorter overall survival in PDAC patients [99]. In addition, a higher NLR and PLR at diagnosis have been associated with R0 resectability and have been inversely associated with nodal status [100].

Notably, the combination of NLR ≥ 1.69 and CA 19-9 ≥ 107.95 U/mL has been shown to be an effective prognostic marker of 100% two-year mortality in PDAC patients with recurrent disease [101]. Recently, the C-reactive protein/lymphocyte ratio has been associated with poor survival at values higher than 1.8. Furthermore, a ratio over 1.8 has been recognized as an independent risk factor of death in stages II, III, and IV [102].

Lastly, patients with PDAC resection and a lymphocyte/monocyte (LMR) ratio ≥ 2.8 were found to have an almost twofold higher overall survival rate at one year compared to patients with an LMR < 2.8 (66.2% vs. 36.1%, respectively; *p* = 0.015) [103].

### 2.12. Human Equilibrative Nucleoside Transporter 1 (hENT1): Prognostic and Predictive Role

Human equilibrative nucleoside transporter (hENT1) is a transmembrane protein that mediates the intracellular uptake of nucleosides or nucleoside-like drugs, including the anti-neoplastic drug gemcitabine. PDAC tumors abundantly express hENT1, and it has been investigated as a potential predictive biomarker of the response to gemcitabine based treatment. In a subanalysis of the ESPAC-3 trial population, which was a comparative study between gemcitabine and 5-fluorouracil based chemotherapy in PDAC patients, hENT1 expression was identified as a predictive biomarker of the response to gemcitabine without any hENT-1-dependent difference observed in the fluorouracil group [104]. Aoyama and colleagues reported that patients with high hENT1 expression in PDAC tissue and treated with curative resection and adjuvant gemcitabine had a higher overall survival at five years (high hENT1: 20.6% vs. low hENT1: 8.9%; *p* = 0.019) and disease-free survival rates at three years (high hENT1: 23.8% vs. low hENT1: 9.4%; *p* = 0.024) post-procedure [105]. In their meta-analysis, Bird et al. showed benefits in both disease-free (hazard ratio: 0.58; 95% CI: 0.42–0.79) and overall survival (hazard ratio: 0.52; 95% CI: 0.38–0.72) in PDAC patients with high hENT1 expression and adjuvant gemcitabine chemotherapy after PDAC resection [106]. In PDAC patients treated with resection and adjuvant S-1, which is a newer oral inhibitor of dihydropyrimidine dehydrogenase, a high hENT1 expression has been associated with a significantly lower median overall survival (30.9 vs. 58.0; hazard ratio: 1.75) [107,108]. In vitro experiments have confirmed the association between hENT1 and gemcitabine effectiveness in PDAC treatment and proposed an inhibitory effect on HIF-1α mRNA levels and protein expression as one of the mechanisms [109]. HIF-1α promotes the survival of cancer cells in hypoxic conditions through the upregulation of glycolysis and has been previously associated with the resistance of malignant cells to chemotherapeutic agents [110]. Lastly, the heterogeneity in the quantification of hENT1 protein expression between various methods has resulted in the evaluation of the hENT1 mRNA level as an effective alternative biomarker [111,112].

### 2.13. Human Concentrative Nucleoside Transporters 1 and 3 (hCNT1 and hCNT3): Prognostic and Predictive Role

The human concentrative nucleoside transporters 1 and 3 (hCNT1 and hCNT3) significantly contribute, in addition to hENT1, to the intracellular uptake of gemcitabine. Previously, in vitro studies revealed that reduced hCNT1 expression is responsible for the resistance of PDAC cells to gemcitabine, while the induction of hCNT1 greatly improved the intracellular gemcitabine uptake [113]. In another in vitro study by Paproski et al., it was shown that the transfection of resistant PDAC cells with hCNT3 cDNA resulted in an increased gemcitabine uptake [114]. Notably, Hesler and colleagues showed that decreased hCNT3 expression in PDAC tumors is dependent on interactions with the microenvironment, specifically the pancreatic stellate cells, which secrete the cysteine-rich angiogenic inducer 61 (CYR61) protein after TGF-β signaling. CYR61 negatively regulates hCNT3 expression in PDAC cells [115]. A splice variant of the hCNT1 RNA, named hCNT1-IR, was recently reported to be overexpressed in PDAC [116]. Interestingly, the chemoresistance of PDAC tumors associated with the tyrosine kinase receptor erythroblastic leukemia viral oncogene homolog 2 (ErbB-2, HER2/neu) or the MUC-4 mucin have been partially attributed to hCNT1 and hCNT3 underexpression [117,118]. In terms of prognosis, a high expression of hCNT3 in PDAC patients treated with gemcitabine/radiation has been associated with a prolonged overall survival, and patients with both high hENT1 and high hCNT3 were found to have an improved median overall survival (94.8 months) compared to patients with the increased expression of only one biomarker (18.7 months) [119].

### 2.14. BRCA1 and BRCA2: Prognostic and Predictive Role

BRCA (breast cancer type 1 and type 2 susceptibility proteins—BRCA1 and BRCA2) inactivating mutations have been previously identified as late molecular events in the pathogenesis of PDAC and have been proposed as predictive and prognostic biomarkers in patients with PDAC [120,121,122,123]. BRCA tumor suppressor proteins participate in the process of DNA double-strand break repair through homologous recombination [124]. Variants in BRCA2 are the most common high-penetrant genetic factors associated with PDAC [125]. Patients with germline BRCA2 mutations are at an up-to 10-fold higher risk of PDAC than the general population [120]. BRCA1 mutations are less common than BRCA2 mutations in patients with familial PDAC, and patients with germline BRCA1 mutations are at a three-fold higher risk of PDAC development [126].

The DNA repair defect of cells with BRCA mutations has been exploited to develop drugs that result in detrimental DNA damage, cell cycle arrest, and apoptosis. The inhibition of the poly-ADP-ribose polymerase (PARP) enzyme results in the development of DNA lesions that are normally repaired by the BRCA-dependent homologous recombination mechanism [127]. The POLO randomized clinical trial investigated the effect of olaparib (PARP inhibitor) on the survival of BRCA1- or BRCA2-positive PDAC patients with metastatic disease. It was shown that olaparib prolongs the progression free survival in patients with germline BRCA mutations (olaparib: 7.4 months vs. placebo: 3.8 months; *p* = 0.004), whereas there was no benefit in terms of overall survival [128]. 

Wattenberg and colleagues recently showed that PDAC patients with BRCA germline mutations have a better objective response rate to platinum-based chemotherapy compared with mutation-negative patients (58% vs. 21% respectively; *p* = 0.0022) [129]. Similarly, in a retrospective study investigating 61 patients with borderline resectable PDAC, BRCA germline mutation carriers receiving neoadjuvant FOLFIRINOX had a higher pathologic complete response rate (44.4%) and overall survival compared to controls (10%) [130].

## 3. Hypoxia as the Hallmark of Pancreatic Cancer Pathogenesis

There is an emerging line of evidence suggesting that hypoxia plays a crucial role in the process of glycolytic metabolism and epithelial–mesenchymal transition (EMT), which collaboratively promote the development and progression of pancreatic cancer [132,133]. The AKT/HIF-1α pathway seems to be involved in this process since a) HIF-1α is a transcription factor involved in tumor metabolic reprogramming in response to hypoxia and b) PI3K/Akt is a frequently activated signaling pathway in pancreatic cancer that is involved in tumor metabolism, malignant transformation, and EMT [134]. A recent study showed that the depletion of HIF-1α impairs the expression of EMT-related markers and the migratory and invasive abilities induced by cancer susceptibility candidate 9 (CASC9). Of interest, CASC9 could promote the activation of AKT, which would be consistent with the observations of enhanced characteristics of EMT in pancreatic cancer cells. Additionally, the inhibition of AKT mitigates these enhanced effects, suggesting that the activation of AKT mediates CASC9-induced glycolysis and EMT in pancreatic cancer. In the same setting, AKT inhibition downregulates the expression of HIF-1α induced by CASC9 [135,136]. The clinical importance of these findings is paramount since the hypoxic pancreatic cancer microenvironment is resistant to chemotherapy and radiation, and it facilitates metastatic behavior [133,137]. 

## 4. Framing Proteomic Work in Pancreatic Cancer

There is an unmet need to develop new biomarkers to guide the diagnosis of pancreatic cancer at early stages, facilitate the differential diagnosis between benign and malignant disease, and guide therapeutic management. Obtaining and translating proteomic findings into clinical practice is an evolving scientific field. Proteins can be detected in body fluids and tissues including serum, cell lines, pancreatic tissue or fluid, and saliva [138]. Mass spectrometry revolutionized proteomic research because it facilitated the biological processing of proteins (intact or enzymatically digested) and the isolation of the proteome. These separations are based on gel chromatography or liquid chromatography [139]. In pancreatic cancer, post-translational modifications of tumor-originating proteins such as the N-glycosylation of glycoproteins have been have been observed to differ between malignant and benign diseases [140]. Similarly, the expression of histone modifications is remarkably different in pancreatic cancer patients compared to healthy controls and is associated with poor outcomes in patients undergoing pancreatic resection [141]. The most attractive strategy to detect pancreatic cancer is via the proteomic analysis of blood due to its availability, accessibility, and ability to reflect and capture dynamic changes in human homeostasis. However, cell lines also have the potential to be used as sources of proteomic biomarkers in pancreatic cancer, with a caveat being that isolated cells are not always representative of total tumor biology. Biomarkers such as SDF4, perlecan, CD9, the fibronectin receptor, and apoE have been identified and validated in primary pancreatic cancer patients, while S100A6 has been associated with metastatic disease [142]. Primary and metastatic cell lines of pancreatic cancer have a remarkably different protein profiles including different expressions of collagens, integrins, galectins, and cadherins that are functionally related to cell motility and adhesion [143]. These findings are of paramount clinical importance because metastasis is the most important cause of death in pancreatic cancer patients. The stromal compartment is another major determinant of the biology of pancreatic cancer; thus, variations of proteomic expression in stromal cells have been shown to be correlated with metastatic nodal disease and poor prognosis [144,145]. The proteomic analysis of pancreatic fluid can also be very informative due to its unique composition. Proteins such as plasminogen, fibrinogen β-chain, caldecrin, neural cell adhesion molecule L1 kallikrein 1 (KLK1), IGFBP2, lithostathine 1, pancreatic secretory granule membrane major glycoprotein (GP2), tumor-associated trypsin inhibitor (SPINK1), pancreatitis-associated protein 1 (PAP1), pancreatic ribonuclease (RNASE1), matrix metalloproteinase-9 (MMP9), oncogene DJ1 (PARK7), alpha-1B–glycoprotein (A1BG), and T-cell receptor beta chain (TCRB) have been shown to be expressed in patients with cancerous pancreatic lesions [146].

## 5. Challenges and Future Directions

Despite the remarkable progress in terms of technology, the nature and biology of pancreatic cancer is not fully understood by clinicians and scientists. The difficulty in identifying biomarkers for the early diagnosis of the disease is mostly based on the fact that despite the identification of a large number and wide spectrum of potential proteomic markers, very few targets have been shown to have a solid scientific role and clinical utility. The identification of an ideal surrogate marker that can provide appropriate accuracy for the screening of pancreatic cancer in the general population is challenging due to the low prevalence of the disease, as well as the absence of specific symptoms (if any) at early disease stages that could trigger a diagnostic workup. Moreover, other organ-specific diseases, such as chronic pancreatitis, pancreatic duct obstruction, jaundice, and diabetes, may confound the performance of a protein biomarker in pancreatic cancer detection. Composite markers or scores, focusing on multi-omics approaches [147] with the concomitant application of computational biology and artificial intelligence principles, can enhance detection accuracy and provide a better robustness. These panels of markers can be initially tested in high-risk patients (smokers, chronic pancreatitis, mucinous neoplasms of the pancreas, and genetic predisposition) before being validated in lower risk patients [148,149,150]. This strategy might provide better efficiency and economic effects for the early detection of pancreatic cancer.

## 6. Conclusions

Pancreatic ductal adenocarcinoma is the most common pancreatic malignancy and presents with scarce symptoms, and its lack of accurate tests results in advanced stage diagnoses and poor prognoses. Multiple biomarkers, identified in both biopsy tissue specimens and plasma, have been investigated as potential diagnostic, prognostic, and predictive biomarkers of pancreatic ductal adenocarcinoma. Despite the discovery of multiple agents, most of them have demonstrated moderate sensitivity and/or specificity and are far from being considered as screening tests. More efficient non-invasive biomarkers are needed to facilitate early-stage diagnosis and interventions. Diagnostic panels that combine known biomarkers seem to be promising cost-effective and time-efficient alternatives to the discovery of newer biomarkers. These diagnostic panels could combine biomarkers derived from the same specimen (i.e., body fluid or tissue) or combine biomarkers derived from different specimens, including pancreatic tissue, serum, peripheral blood, pancreatic secretions, saliva, and urine. Lastly, multi-institutional collaborations that provide adequate sample sizes are essential in the evaluation of any novel biomarker or diagnostic panel.

## Figures and Tables

**Figure 1 cancers-13-01071-f001:**
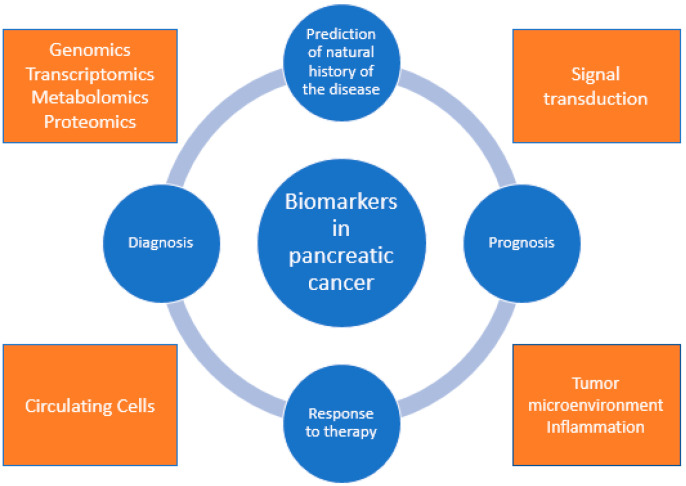
Illustration of the spectrum and variability of biomarkers involved in the diagnosis, prognosis, and prediction of pancreatic cancer.

**Table 1 cancers-13-01071-t001:** Summary of the main roles of commonly used biomarkers. CA 19-9: carbohydrate antigen 19-9; MUC5AC: mucin 5AC; SMAD4: mothers against decapentaplegic homolog 4; DPC4: deleted in pancreatic cancer; hENT1: human equilibrative nucleoside transporter 1; hCNT: human concentrative nucleoside transporter; BRCA: breast cancer susceptibility protein.

Biomarker	Diagnostic	Prognostic	Predictive	Clinical Specimen	References
CA 19-9	X	X		Serum	[9,10,11,12,13,14,15,16,17,18,19,20,21,22,23,24]
microRNAs	X	X		Pancreatic tissue, pancreatic fluid, and urine	[25,26,27,28,29]
DNA methylation	X	X		Pancreatic fluid, endoscopic brush samples, and serum (cfDNA)	[30,31,32,33,34,35,36]
Mismatch repair genes/microsatellite instability	X		X	Pancreatic tissue	[37,38,39,40,41,42,43,44,45,46,47]
KRAS	X	X		Pancreatic tissue and serum (cfDNA)	[48,49,50,51,52,53,54,55,56,57]
Exosomes	X	X		Serum and saliva	[58,59,60,61,62,63,64,65,66,67,68,69,70,71,72]
Circulating tumor cells	X	X		Peripheral blood	[73,74,75,76,77]
PAM4/MUC5AC	X			Pancreatic tissue	[78,79,80,81]
Osteopontin	X	X		Serum	[82,83,84,85]
SMAD4/DPC4	X	X	X	Pancreatic tissue	[86,87,88,89,90,91,92,93,94,95,96]
Immune response and inflammatory markers		X	X	Serum and blood	[97,98,99,100,101,102,103]
hENT1		X	X	Pancreatic tissue	[104,105,106,107,108,109,110,111,112]
hCNT1/hCNT3		X	X	Pancreatic tissue	[113,114,115,116,117,118,119]
BRCA1/BRCA2		X	X	Pancreatic tissue	[120,121,122,123,124,125,126,127,128,129,130]

## Data Availability

Not applicable.

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
