# Peer review of "Diagnostic, Predictive and Prognostic Molecular Biomarkers in Pancreatic Cancer: An Overview for Clinicians"

_cancers, 2021, doi:10.3390/cancers13051071_

Round 1
Reviewer 1 Report
In this manuscript, Giannis and colleagues review the diagnostic, predicitive and prognostic molecular biomarkers in pancreatic cancer. Overall, the article is well written. However, proteomics - an important filed in pancreatic cancer biomarker development was left out without any discussions. This review does not feel complete without the inclusion of the works done in proteomics. Below are some suggestions could improve this review.
Major points
- Please include a section to overview the proteomic works that have been done in applying proteomics to discover or validate pancreatic cancer biomarkers in tissue, plasma/serum, pancreatic juice, cystic fluid or other bodily fluids. Proteins are functional biomolecules involved in all disease stages and have been the major targets in clinical testing. In addition, protein glycosylation is also an important target in pancreatic cancer biomarker development. In fact CA19-9 is a glycosylation assay. It is essential to include the proteomic works done in this field and discuss the emerging proteomic discovered biomarkers in this review.
- In addition to overview the existing biomarkers, the authors should also discuss the current status, challenges and future direction in this field.
Minor points
- In the introduction, 5 year survival rates in different stages of the disease-after diagnosis could be described. Then, the importance of early diagnosis can be elaborated.
- Figure1: please revise Figure 1 to include proteomics.
- Table 1: for each biomarker listed, please include the suitable clinical specimens for testing.
- Line 64: Spell out CEA
- Certain biomarkers have been reported as early or late molecular changes. So please indicate them in the relevant sections with literatures. For examples: KRAS: early molecular changes.SMAD4 and BRCA: Later stage inactivation
Author Response
REVIEWER 1
In this manuscript, Giannis and colleagues review the diagnostic, predicitive and prognostic molecular biomarkers in pancreatic cancer. Overall, the article is well written. However, proteomics - an important filed in pancreatic cancer biomarker development was left out without any discussions. This review does not feel complete without the inclusion of the works done in proteomics. Below are some suggestions could improve this review.
Major points
- Please include a section to overview the proteomic works that have been done in applying proteomics to discover or validate pancreatic cancer biomarkers in tissue, plasma/serum, pancreatic juice, cystic fluid or other bodily fluids. Proteins are functional biomolecules involved in all disease stages and have been the major targets in clinical testing. In addition, protein glycosylation is also an important target in pancreatic cancer biomarker development. In fact CA19-9 is a glycosylation assay. It is essential to include the proteomic works done in this field and discuss the emerging proteomic discovered biomarkers in this review.
Thank you for your comment. We agree that research on proteomic biomarkers in pancreatic cancer is a rapidly evolving field. We added a paragraph framing the work being currently ongoing in the field.
- In addition to overview the existing biomarkers, the authors should also discuss the current status, challenges and future direction in this field.
Thank you for the comment. We discussed the challenges and the future directions in the field.
Minor points
- In the introduction, 5 year survival rates in different stages of the disease-after diagnosis could be described. Then, the importance of early diagnosis can be elaborated.
We thank the reviewer for their insightful comment. We have added the survival rates across different stages to stress the importance of early diagnosis and its association with survival.
- Figure1: please revise Figure 1 to include proteomics.
We have modified our figure to include proteomics as per the reviewer’s suggestion
- Table 1: for each biomarker listed, please include the suitable clinical specimens for testing.
We have modified table 1 to include suitable clinical specimens for each biomarker
- Line 64: Spell out CEA
Thank you for noticing this, we have modified the text accordingly.
- Certain biomarkers have been reported as early or late molecular changes. So please indicate them in the relevant sections with literatures. For examples: KRAS: early molecular changes.SMAD4 and BRCA: Later stage inactivation
We thank the reviewer for this suggestion. We have modified our KRAS, SMAD, and BRCA accordingly and have added relevant references to support these modifications.
Reviewer 2 Report
The review was aimed to provide non invasive PDAC biomarkers for clinician based on the reference knowledge. Various type of molecules from variety of body fluids were introduced as diagnostic, prognostic and predictive biomarkers.
Although each of the topic was interesting and useful, it was diverse in detail and not easy to understand what liquid biopsy approach would provide what clinical outcome. Organizing the expanded Table 1 or separate Table to display the biomarker category, what molecule, from what body fluid and their clinical assessment eg. sensitivity and specificity results may help readers have an idea of the clinical utility.
The authors discussed about the possibility of diagnostic panels for the future direction (Line 383-385). It would be interesting to give the opinion whether the panel could be from same molecular spices or possibly combination of different types of molecules for example cfDNA, microRNA and protein the authors introduced.
Author Response
REVIEWER 2
The review was aimed to provide non invasive PDAC biomarkers for clinician based on the reference knowledge. Various type of molecules from variety of body fluids were introduced as diagnostic, prognostic and predictive biomarkers.
Although each of the topic was interesting and useful, it was diverse in detail and not easy to understand what liquid biopsy approach would provide what clinical outcome. Organizing the expanded Table 1 or separate Table to display the biomarker category, what molecule, from what body fluid and their clinical assessment eg. sensitivity and specificity results may help readers have an idea of the clinical utility.
Thank you for your insightful comment. We have now modified our table adding a column that includes the specimen/body fluid that is suitable for each biomarker. However, we decided to not include sensitivity and specificity, which have been extensively discussed within text. The reason for this decision is the high heterogeneity between the populations that have been used to investigate the value of biomarkers (i.e. patients with PDAC diagnosis at different stages (and markers with prognostic and/or diagnostic utility) and healthy populations (diagnostic utility)). This limitation makes the applicability of any findings individualized and difficult to be applied and interpreted blindly in all patients with PDAC. To our perspective, raw numbers of sensitivity and specificity in the table would be confusing, rather than helpful to the reader.
The authors discussed about the possibility of diagnostic panels for the future direction (Line 383-385). It would be interesting to give the opinion whether the panel could be from same molecular spices or possibly combination of different types of molecules for example cfDNA, microRNA and protein the authors introduced.
We have added the following statement to clarify that diagnostic panels may combine biomarkers derived from the same specimen or biomarkers derived from various specimens, including body fluids or tissue:
“These diagnostic panels could combine biomarkers derived from the same specimen (i.e. body fluid or tissue) or combine biomarkers derived from different specimens, including pancreatic tissue, serum, peripheral blood,, pancreatic secretions, saliva, and urine.”
Reviewer 3 Report
This is a very well written and presented review on prognostic molecular biomarkers in pancreatic cancer with 120+ references most of which to very recent literature.
The work is interesting and useful for the journal readership and can be accepted with minor editorial changes.
Decision: Accept.
Author Response
REVIEWER 3
This is a very well written and presented review on prognostic molecular biomarkers in pancreatic cancer with 120+ references most of which to very recent literature.
The work is interesting and useful for the journal readership and can be accepted with minor editorial changes.
Decision: Accept.
Thank you for your comments.
Reviewer 4 Report
In the current review Giannis et al., authors have summarized role of different biomarkers in diagnostic and management of pancreatic cancer. Pancreatic ductal adenocarcinoma is one of the most lethal and asymptomatic cancer which only becomes apparent after metastasis. The review is written well, precisely, and at a substantial depth. Authors have described several parameters which can be utilized efficiently in early diagnosis of PDAC.
I do not have major suggestions and changes. However, it will be helpful if authors can add the another column in Table 1 providing the references for the mentioned biomarkers. This will improve the usefulness of the Table. Additionally, different biomolecules involved in cellular stress/organoid stress, epithelial to mesenchymal transition (EMT), cell cycle regulating proteins, stress response proteins can be described to elaborate the scope of the review.
Author Response
REVIEWER 4
In the current review Giannis et al., authors have summarized role of different biomarkers in diagnostic and management of pancreatic cancer. Pancreatic ductal adenocarcinoma is one of the most lethal and asymptomatic cancer which only becomes apparent after metastasis. The review is written well, precisely, and at a substantial depth. Authors have described several parameters which can be utilized efficiently in early diagnosis of PDAC.
I do not have major suggestions and changes. However, it will be helpful if authors can add the another column in Table 1 providing the references for the mentioned biomarkers. This will improve the usefulness of the Table.
Thank you. Table 1 has been revised according to the reviewer’s recommendation
Additionally, different biomolecules involved in cellular stress/organoid stress, epithelial to mesenchymal transition (EMT), cell cycle regulating proteins, stress response proteins can be described to elaborate the scope of the review.
Thank you. We added a paragraph highlighting the role of hypoxia and AKT pathway in pancreatic cancer pathogenesis and the clinical implications of this positive feedback loop
Round 2
Reviewer 1 Report
No additional comments.